# Design and Synthesis of Novel Raman Reporters for Bioorthogonal SERS Nanoprobes Engineering

**DOI:** 10.3390/ijms23105573

**Published:** 2022-05-16

**Authors:** Caterina Dallari, Riccardo Innocenti, Elena Lenci, Andrea Trabocchi, Francesco Saverio Pavone, Caterina Credi

**Affiliations:** 1European Laboratory for Non-Linear Spectroscopy, University of Florence, 50019 Sesto Fiorentino, Italy; innocenti@lens.unifi.it (R.I.); francesco.pavone@unifi.it (F.S.P.); credi@lens.unifi.it (C.C.); 2Department of Physics, University of Florence, 50019 Sesto Fiorentino, Italy; 3Department of Chemistry, University of Florence, 50019 Sesto Fiorentino, Italy; elena.lenci@unifi.it (E.L.); andrea.trabocchi@unifi.it (A.T.); 4National Institute of Optics, National Research Council, 50019 Sesto Fiorentino, Italy

**Keywords:** bioorthogonal molecules, organic synthesis, gold nanostars, Raman, surface-enhanced Raman spectroscopy

## Abstract

Surface-enhanced Raman spectroscopy (SERS) exploiting Raman reporter-labeled nanoparticles (RR@NPs) represents a powerful tool for the improvement of optical bio-assays due to RRs’ narrow peaks, SERS high sensitivity, and potential for multiplexing. In the present work, starting from low-cost and highly available raw materials such as cysteamine and substituted benzoic acids, novel bioorthogonal RRs, characterized by strong signal (10^3^ counts with FWHM < 15 cm^−1^) in the biological Raman-silent region (>2000 cm^−1^), RRs are synthesized by implementing a versatile, modular, and straightforward method with high yields and requiring three steps lasting 18 h, thus overcoming the limitations of current reported procedures. The resulting RRs’ chemical structure has SH-pendant groups exploited for covalent conjugation to high anisotropic gold-NPs. RR@NPs constructs work as SERS nanoprobes demonstrating high colloidal stability while retaining NPs’ physical and vibrational properties, with a limit of detection down to 60 pM. RR@NPs constructs expose carboxylic moieties for further self-assembling of biomolecules (such as antibodies), conferring tagging capabilities to the SERS nanoprobes even in heterogeneous samples, as demonstrated with in vitro experiments by transmembrane proteins tagging in cell cultures. Finally, thanks to their non-overlapping spectra, we envision and preliminary prove the possibility of exploiting RR@NPs constructs simultaneously, aiming at improving current SERS-based multiplexing bioassays.

## 1. Introduction

Nowadays, the research in the field of diagnostics and biosensors is more and more focused on the development of novel methods for the molecular screening of biological fluids. As a matter of fact, liquid biopsies have emerged as powerful diagnostic tools not only because of their advantages related to the reduction in invasiveness and risks for the patients but also because it is possible to discriminate between physiological and pathological conditions by precociously identifying analytes (chemicals, biomolecules, cells, etc.) at concentration down to picomolar (pM) [1]. To address this issue, analytical devices must be characterized by high specificity, high sensitivity, quickness, and accuracy. In addition, there is a strong demand for high-throughput analysis enabling to qualitatively or quantitatively reveal multiple analytes within the same reduced volume of the sample [2]. The simultaneous profiling of multiple markers could improve the performances of these analytical devices, and moreover, it is fundamental for those diseases whose early-stage diagnosis relies on the ratio of at least two pathological biomarkers [3]. In this scenario, optical bioassays based on surface-enhanced Raman spectroscopy (SERS) are consolidated as detection methods for the non-destructive molecular screening of biofluids, avoiding sample preparation and possibly affecting their chemical composition and thus the final assay response [4,5]. Taking advantages of surface-localized plasmons phenomena occurring upon metallic nanoparticles (NPs)–light interactions, the Raman signal scattered from target analytes that are adsorbed onto SERS NPs is enhanced (up to 10^6^) with respect to the overall signal from the background biomatrices. Consequently, in order to improve the sensitivity and specificity of the technique, the surface chemistry of NPs can be engineered to increase the affinity toward the biomarkers of interest [6]. Among all reported morphologies, highly branched star-shaped gold particles (AuNSts) are particularly appealing SERS substrates due to a large number of intrinsic hotspots and the high surface-to-volume ratio compared to isotropic spherical NPs [7]. Furthermore, Au surfaces provide an interface for easy covalent conjugation of biomolecules (e.g., antibodies) by thiol-gold chemistry [6,8]. Consolidated gold functionalization strategies can be exploited to further decorate NPs surfaces with Raman reporters (RRs), a special class of compounds whose spectral signature exhibit very narrow bandwidth (~1 ÷ 2 nm) and intense peaks [9,10]. Exactly the RRs represent a powerful alternative to organic dyes, fluorescent probes, and inorganic materials exploited in standard diagnostic methods (e.g., enzyme-linked immunosorbent assay). These present several drawbacks, including autofluorescence, susceptibility to photobleaching, and, among all, spectral overlapping due to their broad linewidth (~50 nm). Furthermore, fluorescent emitters can interfere with biological matrices, as there are limited probes available in the near-infrared region (NIR) where fluorescence emitting from endogenous biological species (proteins, amino acids, cytochrome, phospholipids) is minimized [11,12,13,14]. Conversely, a larger set of NIR-active RRs tags derived from combining Au NPs with molecules characterized by chemical structures emanating different vibrational modes [15]. As a step forward, to overcome cross talking due to RRs’ multiple shifts in the fingerprint region [16,17], a novel class of bioorthogonal RRs has been recently introduced. Biorthogonal RRs are synthesized to show single narrow vibrational peaks in the biological Raman-silent region (1800–2800 cm^−1^), where it is free of background interference and the signals of biospecies are negligible, thus resulting in a higher signal-to-noise ratio. Among the chemical moieties, alkyne, azide, and nitrile groups have been mainly exploited to design background-free nanotags to be anchored to both gold [18,19,20,21] and silver NPs [22,23]. However, reported studies still do not address all the requirements at once, as they rely on long-step chemical procedures with a low yield to obtain molecules either not available for direct covalent absorption on SERS NPs or not providing pendant groups for further conjugation with molecules for biological targeting [20,21]. In the present work, a versatile, modular, and straightforward method is presented to synthesize a novel set of biorthogonal RRs whose chemical structure is rationally engineered for covalent conjugation to gold NPs and to work as optimal SERS nanotags. Starting from commercially available raw materials, the RRs are synthesized to have thiol end-chain groups for one-step covalently binding to NPs as well as to have aromatic rings in the backbone to co-work with PEG as NPs-capping agent. AuNSts morphology and colloidal stability were preserved even upon conjugation to anisotropic spiky star-shaped NPs [24]. RRs NPs loading was optimized to guarantee the highest signal from alkyne- and cyano-derived moieties of the aromatic substituents acting as strong NIR emitting Raman signals with a limit of detection at concentration down to pM. SERS nanotags can provide selective biological tags by simply assembling antibodies on RRs-NPs surfaces decorated with carboxyl groups deriving from PEG molecules [6], thus further highlighting the versatility of the methods. The sensitivity and specificity of the SERS nanoprobes were attested by implementing proof-of-concept experiments in biologically relevant and molecularly complex substrates such as avidin-streptavidin sandwiches and in vitro cell cultures [6]. Finally, the proposed method potentially enables the synthesis of a larger set of tags, hence providing a straightforward method for custom multicolor palette for application in SERS-based multiplexing analyses.

## 2. Results and Discussions

### 2.1. Rational Design and Synthesis of Novel RRs Molecules

New molecular entities acting as bioorthogonal RRs were rationally designed to show (i) high Raman cross section, (ii) strong single narrow vibrational peaks in the biological silent region (1800–2800 cm^−1^), and (iii) proper pending functional groups to be attached to AuNPs avoiding alteration of colloidal stability. To this end, as depicted in the scheme in Figure 1, the RRs are conceived as constituted by two building blocks covalently coupled together: the “NPs anchoring block” and the “bioorthogonal Raman block”. The “NPs anchoring block” should contain at least a thiol end-chain group for NPs covalent conjugation by Au–S bond formation and a second pendant functional group for covalently coupling with the “bioorthogonal Raman block” (Figure 1). With the purpose of developing a very modular, efficient, and versatile synthesis, we selected the amide bond formation as a coupling reaction. To this end, raw cysteamine 1 was identified as “NPs anchoring block”, as the amine functional group was exploited to couple selected benzoic acid derivatives through standard peptide bonds formation. For the bioorthogonal Raman block, benzoic acids were preferred to aliphatic carboxylic acids since the π-electrons of the aromatic rings are characterized by high polarizability leading to a higher Raman scattering cross section [25]. Indeed, our approach enables us to customize a library of bioorthogonal RRs by simply exploring varied benzoic acid derivatives in the synthesis. 

Compared to state-of-the-art studies from the literature focusing on the synthesis of novel bioorthogonal RRs for NPs conjugation [16,26], our synthetic strategy consists of only three low-cost and high-yield steps: (i) protection of the thiol functionality of the NPs anchoring block with triphenylmethanol, (ii) peptide coupling reaction with differently substituted commercially available benzoic acids, (iii) thiol deprotection with trifluoroacetic acid and triisopropylsilane (Figure 2a). It is important to highlight that both compounds composing the backbone of the bioorthogonal RRs are low-cost and commercially available molecules, contrary to commercial RRs generally belonging to the class of bifunctional thiophenols where the simultaneous direct conjugation of the thiol group and the Raman active group to the benzene ring can alter the reactivity as well as the stability of both moieties. Here, as a proof of concept, four probes were successfully obtained by exploiting p-ethynyl-, p-phenylethynyl-, p-cyano- and m-fluoro-, and p-cyano- benzoic acids, all of them characterized by alkyne stretching oscillation with vibrational frequencies above 2000 cm^−1^. As reported in Figure 2a, the thiol functionality of **1** was protected with triphenylmethanol in the presence of trifluoroacetic acid, thus achieving compound **2** in 89% yield, without the need for preventive protection of the amino group, due to the chemoselectivity of this reaction.

Then, benzoic acids were attached to the amine functionality of **2** by using HATU as a selective coupling reagent in the presence of DIPEA and DMF. In this way, different N-(2-mercaptoethyl)benzamide **3a**–**d** were obtained in a range of yields spanning from 54% to 92%, depending on the steric hindrance and the electrophilicity of the carboxylic acid functionality. p-bromo-benzamide derivative **3i** was transformed into the desired p-phenylethynyl-benzamide **3b** by using phenylacetylene in the Sonogashira coupling reaction condition. Finally, the thiol protecting group of **3a**–**d** was removed using a cocktail mixture of trifluoroacetic acid and triisopropylsilane, giving compounds **4a**–**d** as highly stable solids. At the end of the synthesis, solid compounds **4a**–**d**, from here on named RR_1_–RR_4_, were obtained with higher yields compared to reported cases from the literature, soluble at least in ethanol that is crucial for the step of conjugation to NPs.

With the goal of multiplexing, their Raman spectroscopic properties were measured, and as expected, RR_1_–RR_4_ Raman profiles (Figure 2c) were still characterized by overelaborated spectra in the biological fingerprint region (from 1000 to 1800 cm^−1^), with multiple peaks stating for single C-H, C-C, and C-S bonds, carboxylic groups, amide as well as phenyl rings stretching and bending, but also characterized by a single sharp peak in the 2050 cm^−1^ to 2245 cm^−1^ silent region range and highly resolved from each other. Lower frequencies are associated with C≡C stretching, whose smaller reduced mass (µ = 6) produced shorter vibrations. On the other hand, as the reduced mass increases with the C≡N bond (µ = 6.5), peaks are located at higher frequencies. Moreover, the presence of electron-donating or electron-withdrawing substituting groups could influence and shift the vibrational resonance [21].

### 2.2. SERS Bioorthogonal Nanoprobes Engineering and Characterization

SERS bioorthogonal nanoprobes RR@AuNSts were engineered by conjugating RR_1_-RR_4_ to the surfaces of AuNPs. Based on previous results from our groups [27,28], highly branched star-shaped particles (AuNSts) were considered as SERS enhancers due to the convenient overlap between AuNSts plasmonic properties matching the range of excitation wavelength used for Raman analysis of liquid systems as well as for AuNSts morphology with intrinsic “hot spots” [29]. Then, a “one-pot” competitive method was implemented to simultaneously intercalate thiolated polyethylene glycol (PEG) and RRs molecules, both co-working as NPs-capping agents (Figure 3).

To this end, while PEG/NP ratio was kept constant at 500 k to achieve the 100% particle coverage as previously demonstrated [6], different RR/NP ratios were investigated to study the effect of RRs at 5 k, 10 k, 25 k, and 50 k ratios. Optimal PEG/NPs/RRs incubation conditions were those giving brighter bioorthogonal Raman signals while retaining SERS nanotags morphology and avoiding particle aggregation. From this perspective, Raman and UV-Vis absorption spectra were acquired over time for different batches (Figure 4). Experimental data were analyzed as a function of RRs loading in terms of the Raman Intensity Value (RIV) parameter, which quantifies the intensity of the specific bioorthogonal peaks, and DeltA parameter, defined as the percentage of variation of the NPs resonant peak within 7 days. The highest RIVs combined with the lowest DeltAs identify the RR@NPs working window conditions. As shown in the graphs in Figure 4a,b, despite the decrease in the Raman signal when passing from 5 k to 10 k, RIVs for RR_1_ and RR_3_ @AuNSts 10 k did not significantly vary with respect to 5 k (NS), indicating a slight variability between 5 k and 10 k conjugation process. Conversely, in the same range from 5 k to 10 k, RIV for RR_2_@AuNSts significantly decreased (*p* < 0.0001). For RR/NP ratios higher than 10 k RIVs for RR_1_, RR_2_, and RR_3_ dramatically decreased (*p* < 0.0001). Finally, RR_4_@AuNSts showed high signal stability at all RR/NP ratios. With respect to the DeltA parameter, the graphs in Figure 4c,d clearly display that for RR_1_, RR_2_, and RR_3_, colloidal stability was immediately lost after the conjugation processes, as related UV-Vis spectra were characterized by flat, broad, and red-shifted curves, stating for uncontrolled aggregation phenomena. This supports the hypothesis that at higher ratios, there are too many RR molecules adhering to gold and competing with PEG stabilizing agent in decorating NPs surfaces, thus affecting PEG steric-stabilization layer efficacy and causing NPs clustering through irreversible interactions. Finally, in the 5 k–10 k range, improved stability at 7 days was observed for RR_1_, RR_2_, and RR_3_ @AuNSts, while again, negligible changes were measured for RR_4_ @AuNSts at all ratios. Finally, according to experimental data, RR/NP optimal ratios were 10 k for RR_1_ and 5 k for RR_2_, RR_3_, and RR_4_.

TEM micrographs showed that NPs morphology was perfectly retained when implementing the optimized RRs conjugation process, thus guaranteeing that also the resonant peak is retained (Figure 5a). DLS and zeta potential (ζ) measurements further attested that a successful ligand exchange process was achieved since the hydrodynamic diameter (Dh) increased by about 50–60 nm with respect to citrate-capped gold nanostars (whose characteristic Dh was 81.3 ± 1.8 nm, data not shown). At the same time, ζ values decreased by at least 3 mV from −24.2 ± 1.5 mV upon PEG-RRs intercalating for RR@AuNSts conjugates (Figure 5b). 

Finally, to determine the limit of detection (LOD) of the SERS nanoprobes, a concentration series of RR_1–4_@AuNSts aqueous solutions was prepared in a triplicate manner within the 10–0.1 nM range that is considered relevant in clinical practice, also considering that no sample drying was due. As illustrated in Figure 6, the SERS intensity of the RR_1–4_ specific bioorthogonal peaks was plotted against the probe concentration to build the calibration curves. For all the RR@AuNSts, the linear trend was observed almost in the 0.1–1.5 nM range. Saturation was reached at higher values (Appendix A). In the linear region, the calibration curves were fitted, and the error bars are standard deviations from three measurements of each sample. Then, LOD was determined as three times the mean of the standard deviation of blank divided by the slope of the calibration curves, while the standard deviation was calculated considering the error propagation (see Appendix A). It resulted that all SERS bioorthogonal complexes are all characterized by LOD values below 1 nM, with slight differences ranging from 60 pM to 1 nM, a key aspect for their parallel exploitability in a bioassay for multiplexing analysis.

### 2.3. SERS Bioorthogonal Nanoprobes against Biological Relevant Environments

In vitro experiments were performed to demonstrate that bioorthogonal SERS nanoprobes can be further modified to tag specific analytes, even in heterogeneous and molecular complex systems, such as cell cultures. As a matter of fact, we report the first example of bioorthogonal RRs SERS nanoprobes showing simultaneously multiple functionalities as cases from literature are either based on RRs molecules not available for direct covalent absorption on SERS NPs or do not provide pendant groups for further conjugation with molecules for biological targeting. For this experimental study, brighter RR_2_@AuNSts was used for demonstration, and -COOH NPs-pendant moieties from assembled PEG were exploited to conjugate Alexa550-WGA, a fluorescent probe selectively tagging cellular membrane proteins. Incidentally, the implemented bio-functionalization protocol is based on standard peptide bond formation, thus exploitable to conjugate all capturing agents exposing amino functionalities. HEK cell cultures were incubated for 1 h with Alexa550-WGA-RR_2_@AuNSts and then washed thoroughly with PBS buffer to remove unbound SERS probes. Cells incubation with bare RR_2_@AuNSts and free Alexa550-WGA solution were performed as controls. As shown in the confocal images reported in Figure 7a, the fluorescence signal from Alexa550-WGA was observed for cells incubated in the presence of WGA, while no signal was observed when incubating with bare RR_2_@AuNSts. Furthermore, the fluorescence intensity from cells incubated with free Alexa550-WGA solution was comparable with that from cells incubated with Alexa550-WGA-RR_2_@AuNsts (Figure 7b). This attested to the high selectivity and specificity of the SERS constructs and, more importantly, demonstrates that the conjugation to RRs@AuNSts does not interfere in the WGA tagging process of the protein’s membrane. Then, the Alexa550-WGA fluorescence signal was exploited to localize SERS constructs and to acquire Raman spectra from cells incubated with Alexa550-WGA-RR_2_@AuNSts and fixed with paraformaldehyde solution since this would not interfere with the measurements in the biological Raman-silent region. Figure 7c displays representative SERS spectra from different regions in the cell cultures, marked with colored squares in Figure 7a (C states for control, S for sample). As shown, spectra obtained from controls C1, C2, and C3 revealed no peak characteristic for RR_2_. Contrariwise, the SERS spectra clearly revealed the strong narrow peak characteristic of the RR_2_ when focusing on the cell membranes (marks S1–S3). Successful colocalization of fluorescence and Raman signal paves the way for the exploitation of these RRs@AuNSts for immuno-SERS microscopy.

Finally, proof-of-concept experiments were implemented to investigate the possibility of exploiting SERS nanoprobes for multiplexing analysis. To this end, we first verified that characteristic signals from the engineered probes were still distinguishable even upon their mixing. As shown in Figure 8a, the bioorthogonal peaks of each tag could be clearly identified when acquiring SERS spectra of RR_1_@AuNSts, RR_2_@AuNSts, and RR_4_@AuNSts solution. The 3:1:3 mixing volume ratio was selected to have a starting RRs mixture with comparable Raman intensity from each tag. RR_3_@AuNSts was not considered since it was completely hindered by the RR_4_@AuNSts signal (Appendix A). This is due to the broadening of the peaks resulting when moving from raw RRs powder to their corresponding nanoconstructs. Then, the potential multiplexing properties of the RR@AuNSts mixed solution were verified in biologically relevant tests, such as biotin–avidin immunoassays used in standard clinical practices. To this end, RRs@AuNSts exposing biotin moieties (b-RRs@AuNSts) were obtained without affecting SERS nanoprobes properties (Appendix A). Then, biotin-modified polymeric substrates were prepared and further incubated with streptavidin, whose tetrameric structure was exploited for further self-assembling of b-RRs@AuNSts through incubation (Figure 8b). Streptavidin concentration was decreased from 10 µg/mL to 0.1 µg/mL, and Raman-SERS spectra were acquired in random single points of the PDMS assay. For all the concentrations tested, raw spectra presented the RR@AuNSts peaks at 2050 cm^−1^, 2218 cm^−1^, and 2235 cm^−1^, thus attesting that multiplexing behavior, previously tested in solution, was preserved even when implemented on immunoassay. No spurious signal from the background could interfere (Figure 8c), apart from the peak at 2150 cm^−1^ ascribable to a stretching vibration frequency coming from the polymeric device (Appendix A). By plotting the Raman intensity of each bioorthogonal peak, in terms of S/N ratio, against streptavidin concentrations (Figure 8d), the same trend of decreasing Raman intensity with decreasing concentration was successfully observed for all the three probes. Furthermore, RRs characteristic signals were detectable even down to 0.1 ug/mL, thus in accordance with the LODs of the systems. Preliminary results support the potentiality of SERS bioorthogonal tags for background-free multiplex and quantitative detection of liquid samples.

## 3. Materials and Methods

### 3.1. Raw Materials

Gold(III) chloride trihydrate (HAuCl_4_·3H_2_O), trisodium citrate dihydrate (C_6_H_5_O_7_Na_3_·2H_2_O), L-(+)-ascorbic acid (AA), silver nitrate (AgNO_3_), hydrochloric acid (HCl), nitric acid (HNO_3_—70%), Poly(ethylene glycol) 2-mercaptoethyl acetic acid (SH-PEG-COOH, Mn 7500), *N*-(3-Dimethylaminopropyl)-*N*-ethylcarbodiimide hydrochloride (EDC), N-Hydroxysuccinimide (NHS), 1-[Bis(dimethylamino)methylene]-1H-1,2,3-triazolo[4,5-b]pyridinium 3-oxid hexafluorophosphate (HATU), trifluoroacetic acid (TFA), triphenylmethanol, triisopropylsilane (TIPS), 2-thioethylamine, benzoic acids, and all organic solvents were purchased from Merck. Biotinylated antibody, biotinylated albumin serum bovine (BSA), streptavidin, and Alexa550-WGA were purchased from ThermoFisher Scientific. DMEM/F12 medium, FBS, penicillin/streptomycin solution, and Opti-MEM were purchased from ThermoFisher Scientific.

### 3.2. Synthesis of Raman Reporters (RR) Molecules

Experimental procedures and characterization data for *N*-(2-mercaptoethyl)benzamide 3a–d and Raman Reporters 4a–d, as well as copies of ^1^H and ^13^C NMR spectra of all new compounds, are reported in detail in the Appendix A.

### 3.3. Gold-Nanoparticles Synthesis

Gold nanoparticles (Au-NPs) were synthesized by the seeded-growth process described by Yuan et al. [30]. The seed solution of 15 nm nanospheres (NSps) was prepared by adding 1.5 mL of 30 mM HAuCl_4_·3H_2_O (1%) to 48.5 mL of boiling and stirring MilliQ (stirring 7 position, 250 °C). After 10 s, 4.5 mL of 38.8 mM sodium citrate solution was added to the solution. The solution was stirred under heating for 15 min and then stirred without heating for 30 min. For nanostars (NSts) synthesis, 0.083 mL of 30 mM HAuCl_4_·3H_2_O (1% solution) was added to 9.917 mL of MilliQ; 60 μL of 1 M HCl and 100 μL of the NSps solution were added to the solution. Then, 100 μL of 2 mM AgNO_3_ and 50 μL of 0.1 M ascorbic acid were added simultaneously. The solution was stirred for 30–60 s, while its color turned from light red to dark grey (or blue). Immediately afterward, NSts were centrifuged for 20 min at 2500× *g* rpm in 1.5 mL Eppendorf tubes and redispersed in 100 μL of distilled water.

### 3.4. Preparation of RR@AuNSts Complexes

SH-PEG-COOH and RR molecules were exploited in a ligand-exchange process to replace the citrate layer with PEG in citrate-capped nanostars (Cit-NSts). AuNPs solution was diluted to 2 nM, and the same volume of 1 mM SH-PEG-COOH aqueous solution was added together with 2 µL of different concentrations of RR molecules in ethanol. Conditions are reported in the table in Appendix A. The solution was left stirred at RT for at least 2 h, then centrifuged at 25 °C, 2500× *g* rpm for 40 min, and redispersed in milliQ water or PBS for further bioconjugation. The concentration of particles was determined according to a formula reported by the method of De Puig et al. [31].

### 3.5. Preparation of RR@AuNSts Bioconjugates

RR@AuNSts complexes were prepared with 1 nM solution of the previously prepared solution. Incubation was in 0.1 M phosphate saline buffer (PBS) at a pH of 7.4 for 2 h at room temperature after 15 min activation with 0.4 mM EDC and 0.1 mM NHS (final concentration). Then, 0.5 µL of 1 mg/mL stock solution of antibody (Biotinylated-Ab and Alexa550-WGA) was added to 500 µL of 1 nM RR@AuNSts, resulting in a 100-times dilution (1 µg/mL antibodies solution). Samples were then centrifuged for 30 min at 4 °C and 2500 rpm, and the pellet was redispersed in 100 μL of PBS buffer with BSA 0.5% *w/v*. For cell incubation, pellets were dispersed in 100 μL of Leibovitz buffer with 3% FBS and 0.5% BSA.

### 3.6. Preparation of the Proof Immunoassay

To fabricate polymeric devices, PDMS liquid precursor and its curing agent were mixed in a 10:1 *w/w* and poured into the glass mold. Crosslinking of the PDMS was achieved after 12 h at RT and 2 h at 70 °C. Once at RT, the PDMS replica was carefully peeled off from the molds and left again at 70 °C for 2 h to ensure that the PDMS was properly cured at the PDMS/air interface. Then, the surface of the device was functionalized with biotin by incubating with biotinylated-BSA solution (1 mg/mL for 3 min) and further incubating with streptavidin at decreasing concentrations (10 µg/mL, 1 µg/mL, and 0.1 µg/mL for 10 min). The tetrameric structure of streptavidin was exploited to further assemble antibody-conjugated RR@AuNSts (5 nM overnight incubation, 4 °C). After each incubation step, substrates were washed thrice with PBS.

### 3.7. Cells Culture

Human HEK neuroblastoma cells (A.T.C.C. Manassas, VA, USA) were cultured in Dulbecco’s Modified Eagle’s Medium (DMEM) (ThermoFisher Scientific, Waltham, MA, USA) F-12 supplemented with 10% FBS, 1% penicillin/streptomycin solution. Cells were cultured in a humidified chamber at 5% CO_2_ and 37 °C and grown until they reached 90% confluence.

### 3.8. Characterization Techniques

The plasmonic resonant properties of gold nanoparticles colloidal solution before and after functionalization processes were characterized in the range from 400 nm to 850 nm with a UV-Vis-NIR spectrophotometer (Lambda 950 instrument, Perkin Elmer). UV WinLab Software was used to acquire spectra, and data were processed with Origin software. The hydrodynamic dimensions and the polydispersity were acquired by dynamic light scattering (DLS) analysis performed with a Malvern Zetasizer Nano series ZS90. Measurements were performed with a fixed scattering angle of 90°, at 25 °C. Each sample was measured three times, and each measurement consisted of about 30 acquisitions. Cumulating statistics were used to measure the hydrodynamic diameter and polydispersity. Data were then processed with Origin software. The structural features of the NPs were characterized by transmission electron microscopy (TEM, CM 12 PHILIPS).

Living HEK neuroblastoma cells were fixed with 4% paraformaldehyde solution for 10 min at 4 °C, then thoroughly rinsed with PBS three times. The analysis of Alexa550-WGA fluorescence was performed after excitation at 561 nm, using a Nikon C2 laser scanning confocal microscope and a Plan Fluor 100 × 1.49 NA oil immersion objective. A series of optical sections (1024 × 1024 pixels) at the cell median planes was taken and analyzed using ImageJ software. All settings, including pinhole diameter, detector gain, and laser power, were kept constant for each analysis.

Raman spectra were collected with a conventional micro-Raman setup (XploRA PLUS Confocal Raman Microscope, Horiba), consisting of a 785 nm laser (Coherent) and a spectrometer with a focal length of 500 mm, equipped with a 600 lines/mm grating. The incident laser power on the sample was about 20 mW. The scattered light was detected by a CCD camera operating at about 350 K. Raman-SERS spectra were recorded in the wavenumber range of 800–2500 cm**^−1^**, the acquisition time was 3 s, and the measurement was repeated three times for spectral averaging. To avoid spurious signals, calcium fluoride Raman slides (CaF_2_, Crystan) were used as substrates. To extract the Raman signal of interest, fluorescence and background signals were subtracted from the acquired raw spectra through Vancouver Raman Algorithm, dedicated software for automatic autofluorescence background subtraction for Raman spectroscopy [32]. Data were further analyzed with Origin software.

## 4. Conclusions

Herein, we report a versatile, modular, and straightforward method for the high yield synthesis of a set of novel molecules with advanced bioorthogonal Raman properties potentially exploitable for SERS multiplexing bioassays. By combining low-cost and commercially available raw materials, such as cysteamine and differently substituted derivatives of benzoic acids, a series of novel bioorthogonal RRs are obtained and further conjugated to gold nanostars to engineer highly stable SERS nanoprobes retaining optical and physical properties and with LODs in the picomolar clinical range of interest. These background-free RRs@AuNSts can be modified to have bio-tagging specificity for selective targeting of analytes of interest expressed in highly heterogeneous environments such as biological samples. In addition to the stability and to the bio-tagging capability, SERS nanoprobes can be used simultaneously as they retain their bioorthogonal properties of RRs@AuNSts with non-overlapping peaks in the cellular Raman-silent region, thus representing a potential tool to be exploited for parallel detection of more analytes for multiplexed encoding and detection.

## Figures and Tables

**Figure 1 ijms-23-05573-f001:**
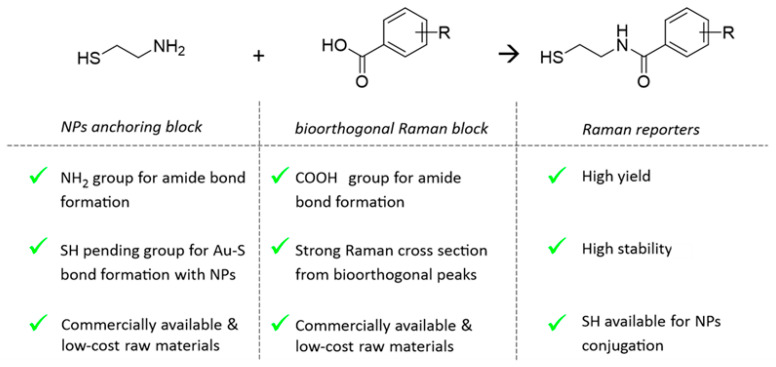
Scheme reporting the key chemical moieties and related advantages for the two compounds constituting the building blocks selected to engineer novel bioorthogonal RRs conjugable to gold NPs.

**Figure 2 ijms-23-05573-f002:**
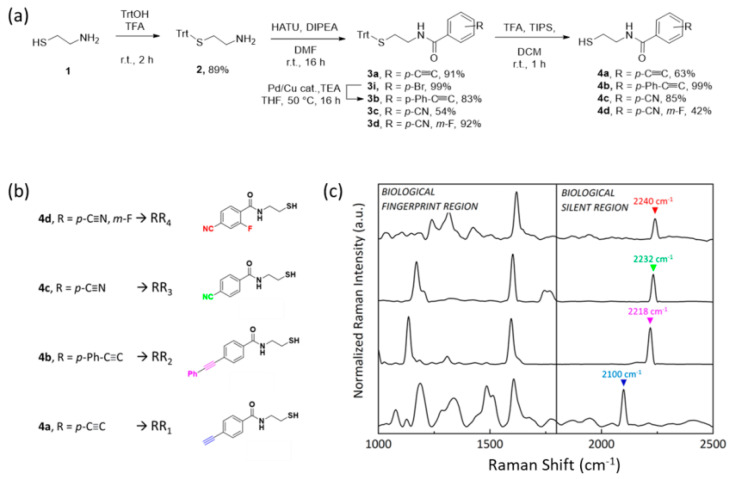
(**a**) General strategy implemented to synthesize bioorthogonal Raman reporters **4a**–**d** from cysteamine 1. (**b**) Chemical structure of the four compounds **4a**–**d** and (**c**) normalized Raman spectra of the **4a**–**d** powder.

**Figure 3 ijms-23-05573-f003:**
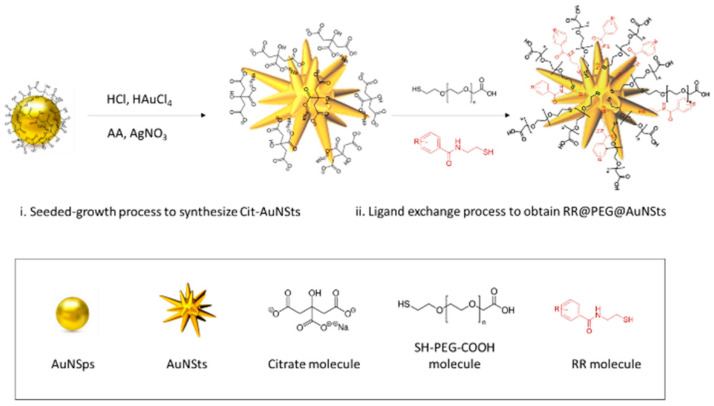
Scheme of the one-pot strategy implemented for SERS nanoprobes engineering.

**Figure 4 ijms-23-05573-f004:**
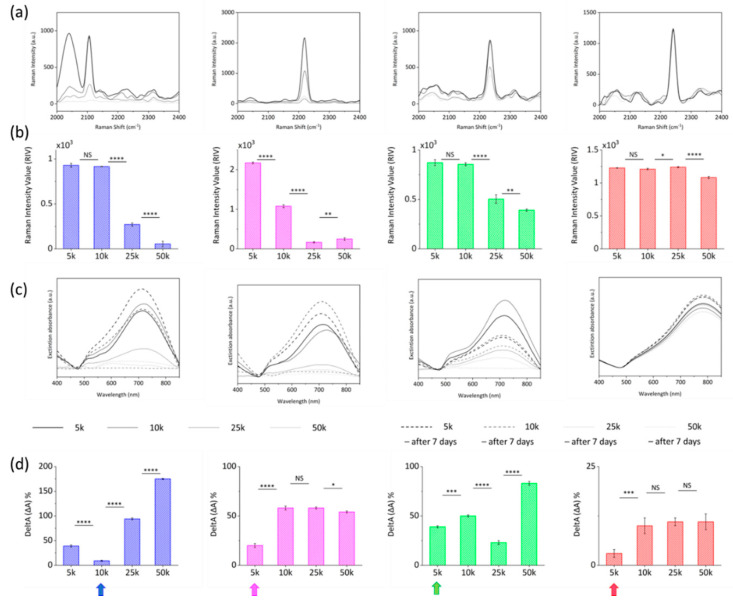
(**a**) Raw Raman-SERS spectra and (**b**) corresponding Raman Intensity Value parameter, acquired for the RR@AuNSts solutions obtained at the different RR/NP ratios investigated (*n* = 3; error bars represent s.d., NS > 0.05; * *p* < 0.05; ** *p* < 0.01; *** *p* < 0.001; **** *p* < 0.0001; an unpaired Student’s *t*-test was performed). (**c**) Extinction spectra of RR@AuNSts solutions at the different RR/NP ratios immediately after conjugation (continue black lines) and after 7 days (dotted lines). (**d**) Corresponding DeltA values parameter (ΔA% = (|At_0_ − At_7days_|/At_0_) × 100) for each RR@AuNSts solution at the different RR/NP ratios (*n* = 3; error bars represent s.d., NS > 0.05; * *p* < 0.05; ** *p* < 0.01; *** *p* < 0.001; **** *p* < 0.0001; an unpaired Student’s *t*-test was performed). For each RRs the final condition is highlighted with a pointing arrow.

**Figure 5 ijms-23-05573-f005:**
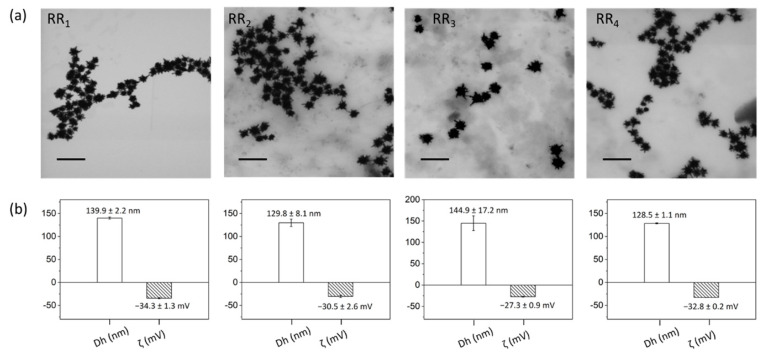
(**a**) TEM images, (**b**) size distribution (Dh), and zeta potential values (ζ) of RR@AuNSts acquired after the implementation of the optimized RRs conjugation. Scalebar is 200 nm in all cases.

**Figure 6 ijms-23-05573-f006:**
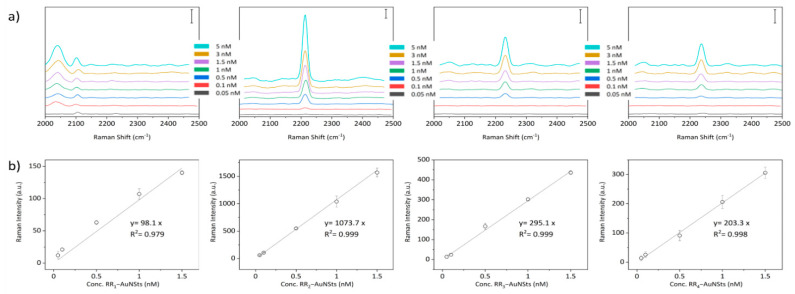
(**a**) Raman-SERS spectra acquired against RR@AuNSts colloidal solutions at decreasing concentrations, ranging from 5 nM to 0.05 nM. Scale bar is 500 counts in all cases. (**b**) Linear concentration dependence of Raman intensity of the characteristic vibrational peak of the different RR@AuNSts nanoprobes (*n* = 3; error bars represent standard deviation. Solid line shows a linear fitting).

**Figure 7 ijms-23-05573-f007:**
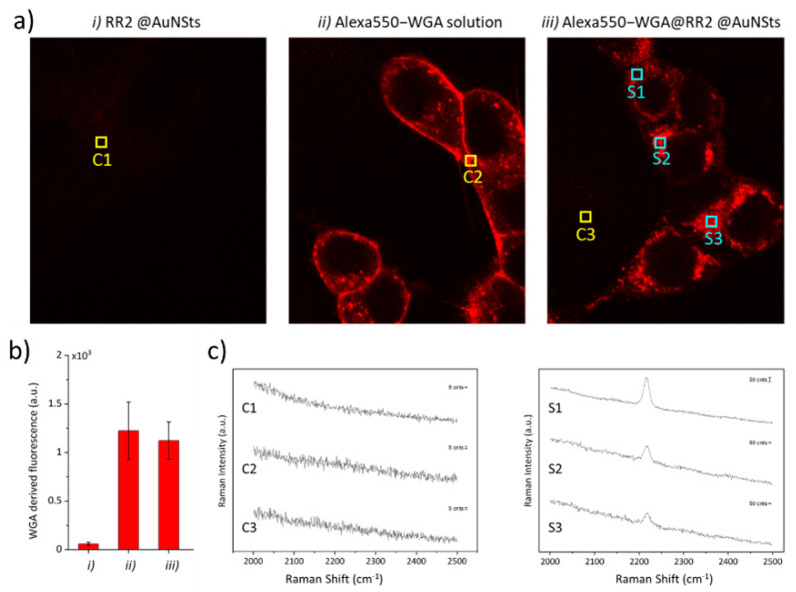
(**a**) Representative confocal microscope images of HEK cells incubated with RR_2_@AuNSts, free WGA solution, and Alexa550-WGA@RR_2_@AuNSts. Scale bar is 100 µm in all cases. (**b**) Histogram showing the quantitative values of WGA fluorescence measured for the three samples after background subtraction. Error bars are standard deviations. (**c**) SERS spectra from random regions in (**a**), indicated by colored squares (C—control; S—sample).

**Figure 8 ijms-23-05573-f008:**
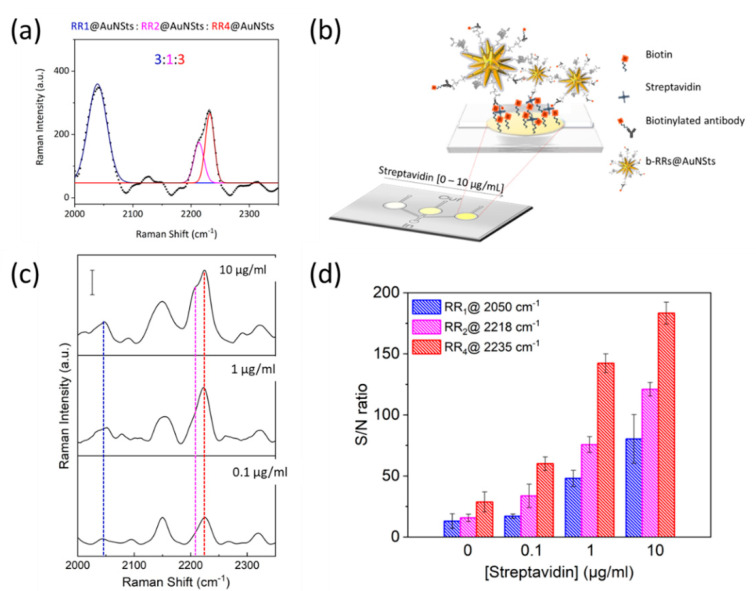
Proof-of-concept experiments implemented to test RRs@AuNSts bioorthogonal capability and potential for multiplexing in standard biotin-avidin assays. (**a**) Multi-component SERS spectra and corresponding deconvolution acquired for RRs@AuNSts mixed solution (black dotted line represents raw data; blue continue line is the fitting for peak of RR_1_; magenta continue line is the fitting for the peak of RR_2_; red continue line is the fitting for the peak of RR_4_; continue black line represents the cumulative fit peaks). (**b**) Schematic representation of the biotin–avidin immunoassay sandwich tests implemented to prove potential RRs multiplexing exploitation. (**c**) Corresponding average SERS spectral response acquired at decreasing concentrations of streptavidin. Scale bar is 50 counts. (**d**) Raman intensity of each bioorthogonal peak showing the same linear trend for all the SERS nanoprobes tested (*n* = 3; error bars represent s. d.).

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
