# Peer review of "Design and Synthesis of Novel Raman Reporters for Bioorthogonal SERS Nanoprobes Engineering"

_ijms, 2022, doi:10.3390/ijms23105573_

Round 1

Reviewer 1 Report

The authors report the synthesis of synthesis of Raman reporters for application in multiplexed Surface Enhanced Raman Spectroscopy assays. The paper is well written and the reported issues have relevance in diagnostics and in biosensing. I believe that the manuscript can be accepted for publication in this Journal after some revisions.

- The synthetic strategy used to produce the compounds 4a-d must be better discussed. In Figure 2a the authors use triphenyl methanol to protect the thiol group. Why the authors did not use triphenylmethyl chloride? Furthermore, are the authors sure that the reaction did not lead to the N-protected molecule? In my opinion, the preventive protection of the amino group should have been considered. The letter (a) in the Figure must be corrected.

- The scale bar of TEM images in Fig. 5 are not clearly visible and should be better highlighted.

Author Response

According to Reviewer #1 “The paper is well written and the reported issues have relevance in diagnostics and in biosensing.” The Reviewer stated that the manuscript deserves publication after some minors that have been answered in the following:

  1. The synthetic strategy used to produce the compounds 4a-d must be better discussed. In Figure 2a the authors use triphenyl methanol to protect the thiol group. Why the authors did not use triphenylmethyl chloride? Furthermore, are the authors sure that the reaction did not lead to the N-protected molecule? In my opinion, the preventive protection of the amino group should have been considered. The letter (a) in the Figure must be corrected.

We thank the reviewer for this suggestion. The use of triphenyl methanol in the presence of an excess of trifluoroacetic acid gave compound 2 in nearly quantitative yield in a chemoselective way. We repeated the reaction three times and we did not observed the formation of the N-protected molecules, as reported also in other papers (Chen, Ying; Kuerbana, Kudelaidi; Wan, Qi; Wang, Ke; Ye, Li; Yu, Zhihui, Molecules, 2020, vol. 25, 3). We did not try the use of triphenylmethyl chloride as reagent, but even in this case, the reaction is reported in the literature and did not require the preventive protection of the amino group (Qu, Yun; Wang, Xinxin; Pei, Zhichao; Pei, Yuxin, ChemMedChem, 2022, 17, 2, E202100548). Accordingly, we revised the discussion of this point in the main text, as follows “without the need of preventive protection of the amino group, thanks to the chemoselectivity of this reaction.” The letters (a), (b), (c) in the Figure have been corrected, as suggested. (Please see the attachment)

  1. The scale bar of TEM images in Fig. 5 are not clearly visible and should be better highlighted.

Following Reviewer’s suggestion, in order to improve the clarity for the readers, the scale bar in TEM images in Fig 5 has been modified. (Please see the attachment)

Reviewer 2 Report

The manuscript by Dallari et al. is an interesting research article about the synthesis and characterization of novel biorthogonal Raman reporters with engineered chemical structure to allow for covalent conjugation to gold nanoparticles and to work as optimal surface-enhanced Raman spectroscopy nanotags.

The topic is of great interest and well fits with the journal scope. The research is well designed and performed, and the whole paper is well organized. This reviewer is suggesting some minor improvements before publication of the paper. Below some comments.

Abstract: Authors should consider the possibility to insert some key data results in the section by reducing the discussion part

Introduction: no specific comments to this section

Results and discussion: authors should consider the possibility to insert some comparison with available systems to enhance the outcome of their research.

Materials and Methods: no specific comments to this section

Conclusion:  no specific comments to this section

Author Response

According to Reviewer #2 “The manuscript by Dallari et al. is an interesting research article about the synthesis and characterization of novel biorthogonal Raman reporters” and “The topic is of great interest and well fits with the journal scope. The research is well designed and performed, and the whole paper is well organized.” Even this second Reviewer stated raised minor comments that have been answered in the following:

 Abstract: Authors should consider the possibility to insert some key data results in the section by reducing the discussion part

 We thank the Reviewer for the suggestion. The abstract has been modified and the revised version is reported in the updated Manuscript.

Results and discussion: authors should consider the possibility to insert some comparison with available systems to enhance the outcome of their research

We thank the Reviewer for the suggestion. In order to enhance the outcome of our research, the main results we obtained have been further highlighted in the Results and Discussion section of revised Manuscript by adding direct comparison with studies from the literature instead of just mentioning them in the Introduction section. Revisions are highlighted in red typing in the revised version of the Manuscript. (Please see the attachment)
